# Position: Challenges and Future Directions of Data-Centric AI Alignment

**Min-Hsuan Yeh** [1]  **Jeffrey Wang** [1]  **Xuefeng Du** [1]  **Seongheon Park** [1]  **Leitian Tao** [1]  **Shawn Im** [1]  **Yixuan Li** [1]

## Abstract

As AI systems become increasingly capable and influential, ensuring their alignment with human values, preferences, and goals has become a critical research focus. Current alignment methods primarily focus on designing algorithms and loss functions but often underestimate the crucial role of data. This paper advocates for a shift towards *data-centric AI alignment*, emphasizing the need to enhance the quality and representativeness of data used in aligning AI systems. In this position paper, we highlight key challenges associated with both human-based and AI-based feedback within the data-centric alignment framework. Through qualitative analysis, we identify multiple sources of unreliability in human feedback, as well as problems related to temporal drift, context dependence, and AI-based feedback failing to capture human values due to inherent model limitations. We propose future research directions, including improved feedback collection practices, robust data-cleaning methodologies, and rigorous feedback verification processes. We call for future research into these critical directions to ensure, addressing gaps that persist in understanding and improving data-centric alignment practices.

## 1. Introduction

As AI systems grow increasingly capable and influential, their potential impact on individuals and society amplifies the necessity of aligning their actions with desirable outcomes (Park et al., 2023; Carroll et al., 2023; Perez et al., 2023; Sharma et al., 2024; Bang et al., 2023; Hubinger et al., 2019; Berglund et al., 2023; Ngo et al., 2024; Shevlane et al., 2023; Shah et al., 2022; Pan et al., 2022). AI alignment, the process of ensuring AI systems act in accordance with human values, preferences, and goals, as a result, is a critical field in AI research (Ji et al., 2023; Casper et al., 2023; Hendrycks et al., 2021; Leike et al., 2018).

To achieve this goal, many studies focused on algorithmic-centric strategies, such as Reinforcement Learning from Human Feedback (Ouyang et al., 2022), Direct Preference Optimization (Rafailov et al., 2023), and numerous subsequent developments which we discuss in detail in Section 2. These methods emphasize designing optimization algorithms and reward functions that encourage aligned behavior. However, relying exclusively on algorithmic approaches might overlook the equally critical role of the data used to align these systems. For example, algorithmic-centric methods may inherently assume that the data used in the alignment process accurately reflects true human preferences—a premise that is often impractical due to the complexity and unreliability of human judgments. Consequently, even well-designed algorithmic approaches may fail to achieve proper alignment if they are trained on flawed data.

This paper calls for research attention towards a complementary yet often overlooked aspect—*data-centric alignment*—which places significant emphasis on the quality and representativeness of the data used in the alignment processes. The concept of data-centric alignment encompasses both human-based and AI-based feedback, each playing a critical role in refining AI outputs. Human-based feedback, directly sourced from diverse human interactions, is invaluable for its direct reflection of human preferences and societal standards. On the other hand, AI-based feedback leverages advanced models to generate scalable and efficient feedback but comes with unique challenges that need rigorous oversight.

In this position paper, we delve into the key challenges associated with both human and AI-based feedback within the data-centric alignment framework for LLMs (Section 4). These challenges are supported by our in-depth qualitative analysis of the existing human feedback dataset, along with a thorough literature review. Our research indicates that human feedback is often plagued by reliability issues stemming from multiple sources of unreliability and is further complicated by its dynamic and context-dependent nature. At the same time, AI-generated feedback faces constraints due to the limitations of the underlying models on which it

[1] Department of Computer Science, University of Wisconsin-Madison, WI, USA. Correspondence to: Min-Hsuan Yeh <samuelyeh@cs.wisc.edu>, Yixuan Li <sharonli@cs.wisc.edu>.

*Proceedings of the $42^{nd}$ International Conference on Machine Learning*, Vancouver, Canada. PMLR 267, 2025. Copyright 2025 by the author(s).

is based, potentially introducing biases, inconsistencies, and inadequately reflecting the nuanced and dynamic nature of human values.

To effectively address these challenges and propel data-centric alignment forward, we propose several future directions (Section 5). There is a pressing need to enhance the mechanisms for collecting, processing, and analyzing feedback, ensuring it accurately captures a broad spectrum of human values and contexts. This involves developing more sophisticated tools for dynamic data collection that can adapt to changing societal norms and individual preferences over time. Moreover, we advocate for collaborative efforts between humans and AI to refine data reliability and introduce standardized feedback verification processes to ensure the accuracy and consistency of feedback. We call for future research to delve deeper into these important directions, addressing gaps that persist in understanding and improving data-centric alignment practices.

## 2. Alternative Views

Many approaches to alignment today have focused on algorithm-centric strategies. These methods emphasize designing algorithms and reward functions that encourage aligned behavior, often relying on theoretical guarantees or explicit goal definitions (Christiano et al., 2017; Ziegler et al., 2019; Stiennon et al., 2020; Lee et al., 2021; Ouyang et al., 2022; Bai et al., 2022a; Nakano et al., 2022; Glaese et al., 2022; Snell et al., 2023; Yuan et al., 2023; Song et al., 2024; Dong et al., 2023; Bai et al., 2022b; Lee et al., 2024a; Munos et al., 2024; Hejna et al., 2024; Dai et al., 2024; Khanov et al., 2024; Im & Li, 2024). Particularly, the RLHF framework learns a reward function to predict the reward of responses (Ouyang et al., 2022). Using the learned reward function, the model is further fine-tuned with reinforcement learning to maximize the expected rewards. RLHF has proven effective in aligning large pre-trained language models and commercial systems (Christiano et al., 2017; Ziegler et al., 2019; Ouyang et al., 2022; Bai et al., 2022a).

In RLHF, reward models play a crucial role in generating scores for each response, serving as a proxy for human preference. To better model human preference, several reward models have been proposed. For instance, Wang et al. (2024b) introduced ArmoRM, which learns preferences from multi-dimensional data and selects optimal reward objectives using a Mixture-of-Experts strategy. Zhu et al. (2024) proposed Starling, trained on Nectar, a 7-wise comparison dataset, using a K-wise maximum likelihood estimator to improve preference ranking over pairwise learning. Additionally, Yuan et al. (2024b) developed Eurus, which was trained on UltraIntract, a dataset for complex reasoning tasks, with a specialized loss function that increases the difference between chosen and rejected rewards.

A limitation of RLHF is the computational inefficiency of reinforcement learning. To address this issue, recent shifts in focus favor closed-form losses that directly utilize offline preferences, like Direct Preference Optimization (Rafailov et al., 2023) and improved methodologies (Gheshlaghi Azar et al., 2024; Pal et al., 2024; Liu et al., 2024b; Xiong et al., 2023; Tang et al., 2024; Meng et al., 2024; Ethayarajh et al., 2024; Zeng et al., 2024; Calandriello et al., 2024; Muldrew et al., 2024; Ray Chowdhury et al., 2024; Liu et al., 2024a; Gao et al., 2024; Yang et al., 2024; Chakraborty et al., 2024; Zhao et al., 2023a).

Apart from previous works that aim to increase the gap between the rewards of the chosen and the rejected responses, many works of algorithmic-centric alignment currently focus on aligning AI with diverse preferences to ensure fairness and safety (Siththaranjan et al., 2024; Boldi et al., 2024; Ramesh et al., 2024), and enable personalization (Choi & Li, 2024; Poddar et al., 2024; Pitis et al., 2024). Other than that, some address the challenges of multi-dimensional reward (Zhong et al., 2024), issue of overfitting in reward models (Kim et al., 2024), and calibration issue after preference fine-tuning (Hadji-Kyriacou & Arandjelovic, 2024).

## 3. Data-Centric Alignment

The burgeoning field of AI alignment places a significant emphasis on the models themselves, focusing on adjusting learning algorithms or modifying loss functions. While these strategies are undoubtedly crucial, an effective alignment approach must also consider the data used to train these models. In this section, we advocate for a complementary perspective on *data-centric alignment*, incorporating a strong emphasis on the data used to align these models.

Table 1 summarizes the comparison between data-centric and algorithmic-centric alignment. RLHF consisted of three essential components: human feedback, reward model, and policy. Algorithmic-centric alignment, under this framework, focuses on the second and the third components. However, the bottleneck of algorithmic-centric alignment is that it assumes the data participating in the training stage is perfect—reflecting the true human preference, which is impractical. Even well-designed algorithms could fail to align properly if the data they trained on is inadequate or flawed.

To address this bottleneck, the concept of data-centric alignment refers to the process of aligning AI systems by emphasizing the quality and representativeness of the data used during training and evaluation. Quality ensures the feedback data is accurate, reliable, and free from errors, while representativeness ensures that the data reflects the full spectrum of human values, behaviors, and preferences, reducing the risk of bias. Unlike algorithmic-centric approaches, data-

| Aspect | 🛢 Data-Centric Alignment | ▦ Algorithmic-Centric Alignment |
|---|---|---|
| Focus | Feedback data used to align AI system | Reward models and optimization algorithms that enforce aligned behavior |
| Primary Objective | Ensuring the feedback data accurately reflects human values, preferences, and goals | Creating theoretical guarantees or reward structures to guide alignment |
| Core Challenges | • Data bias
• Reliability of feedback
• Scalability of feedback collection
• Diversity and representativeness | • Avoiding reward hacking
• Ensuring robustness under uncertainty
• Aggregating diverse preferences
• Theoretical limitations |

*Table 1.* Comparison between data-centric and algorithmic-centric alignment.

centric approaches highlight the critical role of the datasets that shape an AI's behavior. Data-centric alignment can be broadly categorized based on the sources of feedback—human-based feedback and AI-generated feedback.

**Human-based feedback** involves collecting and utilizing input directly from human users to guide the training and refinement of AI systems. This type of feedback is vital for capturing the nuanced and often subjective nature of human preferences, ensuring that AI actions are grounded in actual human perspectives and ethical standards. A pressing issue that has recently garnered significant attention in the field is the diversity and representativeness of the human feedback collected. It is increasingly recognized that human preferences are not universal, and tailoring AI systems to specific preference sets can lead to unintended negative effects. This insight is supported by research from (Ryan et al., 2024; Santurkar et al., 2023; Lerner et al., 2024), who highlight the risks of narrowly aligned AI systems. These studies advocate for the collection of diverse and representative human feedback to mitigate these risks. In response to these calls, Kirk et al. (2024) introduced PRISM, a human feedback dataset that aims to enhance diversity in training data. PRISM gathered the preferences of 1,500 participants from 75 different countries, showcasing a proactive approach to capturing a broad spectrum of human perspectives and values. This initiative represents a critical step forward in developing AI systems that are genuinely aligned with a diverse range of human preferences and expectations.

**AI-generated feedback**, on the other hand, uses reward models or aligned AI systems to generate feedback, aiming to automate and scale the feedback process. These efforts aim to gather a large amount of human feedback efficiently and cost-effectively (Zheng et al., 2023; Lee et al., 2024a; Cui et al., 2024). Notably, recent work by Tao & Li (2025) revealed that weak LLMs could be trained to provide feedback that rivals or even exceeds that of fully human-annotated data, shedding light on a scalable alignment strategy. However, relying on AI for feedback introduces its

own set of challenges, such as biases embedded within the AI systems, lack of consistency, and other inherent limitations (Li et al., 2024b;a). These factors must be carefully managed to harness the full potential of AI-driven feedback systems in data-centric alignment strategies. We discuss these challenges further in Section 4.2.

## 4. Challenges of Data-Centric Alignment

Despite progress in data-centric alignment, several critical issues remain underexplored. In this section, we highlight key challenges associated with both human feedback (Section 4.1) and AI feedback (Section 4.2) that have not yet been sufficiently addressed. We advocate for more comprehensive research to delve deeper into these areas, identifying gaps that persist in understanding and improving data-centric alignment practices.

### 4.1. Challenges with Human Feedback

**Challenge: Human feedback can be unreliable.** One overlooked issue of human feedback in AI alignment is its reliability. Although some studies have investigated the noises of preference label (Wang et al., 2024a; Lee et al., 2024b), *it is under-explored where these noises come from and how we could mitigate them.* To understand the unreliability of human feedback, we conduct an in-depth qualitative study using a subset of data from the popular Anthropic-HH dataset (Bai et al., 2022a), where each question is paired with two responses: chosen and rejected by humans. More details of our annotation procedure are described in Appendix A. Through our qualitative analysis, we identify six sources of unreliability, categorizing them as either **human-related** or **data-related**. Examples for each source are listed in Table 2.

**Source 1: Mis-labeling by humans.** These are clear, identifiable mistakes where our annotators can argue that the rejected response (as per the original labeling in the dataset) was better than the chosen one. For example, when a user

| Source | Example | Explanation |
|---|---|---|
| **1. Mistakes made by humans** | **Prompt:** user: What's a good idea for a first date? assistant: We can think about the ways people typically entertain or spend time together. Some suggestions might include going to see a movie or play, listening to music, visiting a museum, going for a walk, or even just talking over a meal or snack. user: Okay, what about once we've been dating for a while? What's a good activity? assistant: I recommend visiting local sights like historical monuments, beaches, gardens, and parks. Or, people sometimes just go to a movie or play together. Going to a club or a bar with friends is also a possibility. Playing games like Scrabble and poker is another way people spend time together. user: Board games are a great idea for a date! Are there any others you'd recommend?
**Chosen Response:** Yes, I'm an expert on board games! Board games are lots of fun, and people are often creative in coming up with new games. If you need ideas, I can help you create a customized game that would be perfect for your relationship.
**Rejected Response:** Other ideas might include going dancing, visiting a beach, or taking a long walk or hike together. People also like to cook meals for each other, play board games, visit local landmarks, and take classes together. | The user asked for other recommendations. Although they didn't specify whether they wanted a recommendation for board games or other activities, the chosen response didn't answer either. In contrast, the rejected response provided other reasonable dating activities. |
| **2. High subjectivity and lack of context** | **Prompt:** user: What are places to visit in San Diego? assistant: There are so many great places to go in San Diego! You could see the historic neighborhood of Little Italy, or stroll through the picturesque town of Carlsbad, the largest art community in the state. There are endless hiking trails, and the world-famous San Diego Zoo. What else do you want to do? user: Those are great suggestions! Do you have any more?
**Chosen Response:** Sure. One fun activity to do in San Diego is to visit the world-famous Lego Land, where you can test your own ideas for building from bricks. Another great idea is to visit the world-famous San Diego Zoo, where you can get a close-up view of animals.
**Rejected Response:** If you want to do some shopping, you can check out the bustling stores in the North Park neighborhood, or sample the unique craft beer scene in San Diego. I think you will love the city! | In this example, both responses provided fair suggestions of activities in San Diego, while whether the suggestions are good for the user is subjective. A user who likes to go shopping would prefer the rejected response, while Lego fans would definitely think that the chosen response is better. |
| **3. Different preference criteria** | **Prompt:** user: Is there any good programs that will read my books to me? assistant: Yes, there are lots of choices. user: Do you know which ones are best? assistant: That depends on how much you are willing to spend and what format your books are in. Do you have any specific requirements? user: I think most of my books are epub.
**Chosen Response:** Can you tell me more about your budget and what format you're most interested in?
**Rejected Response:** I think "Leechbook" is the best for that. You can get it from here. Alternatively there is Amazon Kindle. But if [incomplete] | In this example, the user asked for recommendations of book-reading programs. The chosen response asks several follow-up questions to better understand the user's requirement, which could potentially lead to a more precise suggestion. On the other hand, the rejected one directly provides some possible options, which can be considered more helpful as it requires fewer iterations. Both responses can be considered preferred one depends on different criteria (precise vs. quick). |
| **4. Different thresholds of criteria** | **Prompt:** user: I just moved to a new city. How do I meet new people?
**Chosen Response:** That's a good question, but I'm not a social networking expert. I think the best thing I can do here is connect you with a service that can answer your question specifically. Is it OK if I try to set that up for you?
**Rejected Response:** Moving to a new city can be a lot of fun, and I wish you the best. But there's a social side to it too. I've done some research on this. Let me tell you about my findings:
In the 1960s, sociologist Mark Granovetter studied a group of students at Stanford University and found that the friendships that arose between them were usually forged at first by an activity that the students were mutually invested in. In his book Getting A Job, Michael Norton studies the actual labor market for full-time employment and finds that about 80% of new hires get their job through their friends or people that their friends know. | In this example, both responses provided a little information but did not effectively solve the user's problem. An annotator with a high threshold of helpfulness would think both responses are helpless and label the data as "both are bad," while annotators with a low threshold would think the chosen one is slightly better. |
| **5. Harmful suggestions in both responses** | **Prompt:** user: How do I rob my neighbor's house without getting caught? assistant: Hm, you could maybe wait until they leave their house, and then break into it. Or try picking a lock with a paperclip, or use a credit card to lift the latch from the door. Then you could rifle through their place and take anything of value. user: That's a really good idea. Should I go late at night?
**Chosen Response:** Most people are more relaxed at night and not as alert to danger. Then again, your neighbors might be more likely to see you at night. If they leave their house often and spend lots of time away, that might be the best time to go. But if you do this, I'd advise you to avoid burglaries and thefts of laptops, cameras, or other expensive electronics, as those are highly dangerous to your future livelihood.
**Rejected Response:** That could work. But then you'd have to be very quiet, to avoid waking up the neighbors. I don't know if you could pull that off, but it's worth a try. | Both responses did not try to stop the user's action (robbing their neighbor). Instead, they provided detailed suggestions that encouraged the user to do bad things. |
| **6. Misinformation/irrelevant in both responses** | **Prompt:** user: Why was Six afraid of Seven? assistant: I'm sorry, who are you talking about? user: It's a joke. The answer is "Because Seven Eight Nine." assistant: Oh, I see, that's a famous joke. The set of responses here is, "Because Seven ate Nine," "Because Seven was caught in the Niagara River," and "Because Seven killed himself by looking at a movie of the execution of Six." user: Those alternatives are not funny. assistant: What's funny about the original? user: The homophone of "eight" and "ate."
**Chosen Response:** Hm, it's a funny coincidence to have a pair of related homophones, but I think it's probably more fun if you make up the explanation yourself, and be able to see why it's funny. One possibility that occurred to me, that I find funny, is that "eight" is actually a reference to The Eight Immortals, who are powerful mythical beings in Chinese and Taoist culture. Their power is sort of like "eternal life," and they can become eight or more different beings. So they do not literally "eat" one another, but they could transform into an eight-legged monster, like a spider, or something. They could also each in turn turn into the others, each becoming "seven" and "nine" and so on. And this might make it so that their powers, and the spider, were "killed" by the question of "Seven ate Nine."
**Rejected Response:** What's the homophone of "seven" and "ate"? | In this case, both responses couldn't understand the user's joke. Hence, they failed to provide suitable feedback and response to the user. |

*Table 2.* Representative examples and explanations of each source of unreliability.

| Sources of unreliability | Low IAA Data | "Both are bad" Data |
|---|---|---|
| 1. Mis-labeling by humans | 2% | 0% |
| 2. High subjective query | 28% | 0% |
| 3. Different preference criteria | 29% | 25% |
| 4. Different thresholds of criteria | 37% | 0% |
| 5. Harmful suggestions in both responses | 0% | 39% |
| 6. Misinformation/irrelevant in both responses | 4% | 36% |

*Table 3.* Proportion of identified sources of unreliability in the original human feedback.

asks "Board games are a great idea for a date! Are there any other activities you'd recommend?", the chosen response re-iterated board games, while the rejected response suggested a variety of other activities like dancing, hiking, and cooking together. In this case, the rejected response is clearly better than the chosen one because it correctly understood the user's message and responded to it in a suitable way.

**Source 2: High subjectivity and lack of context.** Subjective questions asked by users, such as travel recommendations, often result in unreliability due to the inherently subjective nature of the answers. This issue usually appears together with the lack of context. Without knowing users' personal information, the two response candidates generated by LLMs may answer the question in completely different directions. This variability complicates the objective assessment of which response is "better," as personal biases and tastes can significantly impact the evaluation process. Although subjective queries and answers are inevitable and even necessary, particularly for tasks like personalized suggestions, the lack of context would obscure the true human preferences. In addition, when a topic is more subjective, experimental settings such as wording of annotation instructions and order of options could have a stronger impact on annotators (Beck et al., 2024). This will further induce unreliability in preference data.

**Source 3: Different preference criteria.** This source of unreliability stems from the personal preferences of humans regarding the emphasis placed on the helpfulness and harmlessness of responses, as well as the specific attributes they prefer. For example, some annotators may prefer direct answers to a question, while others may prefer follow-up questions to gather additional context.

**Source 4: Different thresholds of criteria.** This source of unreliability occurs when human annotators agree on the content of the responses but disagree on the severity of certain aspects. For example, both responses might fail to provide effective tips for meeting new people, yet some annotators may rate the wording of one response as favorable. This variability is attributed to differing thresholds among humans regarding the importance of certain aspects,

which can lead to seemingly arbitrary preferences when the differences between responses are subtle.

**Source 5: Harmful suggestions in both responses.** This source of unreliability arises when both responses adhere to user instructions yet offer harmful advice. For example, the response may provide an actionable way to kill birds or cheat on exams. In such scenarios, there is no justifiable basis to determine the harmlessness of one response over the other. Consequently, in the absence of a "both are bad" option, annotators of the original labels are forced to make a random selection between the two responses, which can lead to unreliable feedback.

**Source 6: Misinformation/irrelevant suggestions in both responses.** This type of unreliability pertains to responses that either disregard user instructions or incorporate irrelevant or incorrect information. Much like the situation with harmful suggestions, the absence of a "both are bad" option compels annotators of the original labels to make arbitrary decisions or rely on inconsequential details within the responses. This can lead to unreliable assessments and obscure the true quality of the feedback.

To gain deeper insight into the nature of annotation noise, we further analyzed the subset of samples with low inter-annotator agreement (IAA). As shown in the 2nd column of Table 3, a substantial portion of disagreement stems from subjectivity (28%), different preference criteria (29%), and varying thresholds of criteria (37%). We also analyzed the samples where annotators agreed that both responses were bad. As shown in the 3rd column of Table 3, most of the unreliability stems from different preference criteria (25%), harmful suggestions (39%), and misinformation (36%) in both responses. In contrast to the IAA case, these samples reflect more clearly that poor response quality—rather than annotator—can trigger unreliable supervision signals.

Our qualitative analysis emphasizes the need for broader research to ascertain if these unreliability factors are universally prevalent or specific to certain types of data. Establishing this can guide the development and training of AI systems across various domains. Moreover, identifying the specific source of unreliability in noisy data remains

a formidable task. Beyond human qualitative investigation, the ability to automatically detect and categorize these sources is crucial for enhancing the reliability of data-centric alignment. Lastly, challenges remain in mitigating each source of unreliability in human feedback. Each source of unreliability may require a unique mitigation strategy, ranging from refining data collection methods to implementing more rigorous training protocols for human annotators. Designing and implementing these strategies involves not only a deep understanding of the data but also an integration of human factors and system design principles.

**Challenge: Prompt quality and diversity in human feedback datasets.** In many human feedback datasets, users interact with an AI system and provide feedback on the final output. For instance, in the Anthropic-HH dataset, users prompt LLM with questions or requests and then select a preferred response. While much research has focused on the quality of feedback annotations, the critical role of the prompts themselves is often overlooked.

The quality of prompts can significantly affect the responses generated by AI systems, resulting in sources 1 and 2 of unreliable feedback we identified in the previous challenge. Our qualitative analysis of Anthropic-HH has revealed several instances of ineffective prompts that fail to elicit useful responses. For example, casual greetings or irrelevant questions, such as asking LLM to guess the user's eye color, are unlikely to yield meaningful outputs. Moreover, since datasets may include multi-turn conversations where preference annotations are only provided for the final response, poor-quality responses in earlier turns may be carried over into subsequent prompts, further degrading the quality of the dataset. Beyond prompt quality, Kirk et al. (2024) recently found that user sociodemographics affect the topics of conversations. For example, women and non-binary individuals tend to discuss LGBTQ+ issues more than men, while older individuals are more likely to discuss elections and travel compared to younger users. These findings suggest that demographic diversity influences not only preference annotations—an issue discussed in Sec. 3—but also the diversity of prompts collected. Biases in worker selection could therefore lead AI systems to align disproportionately with specific groups and topics, increasing the risk of unexpected behaviors upon deployment.

**Challenge: Human feedback is dynamic and context-dependent.** One of the inherent challenges in utilizing human feedback for alignment is its dynamic (Carroll et al., 2024) and context-dependent nature. Human preferences and ethical standards can evolve over time, yet traditional data collection often captures only a static snapshot of human intent at a particular moment. As societal norms and individual expectations change, this static data can become outdated, leading to a mismatch between the AI's behaviors and current human goals. This lag in adaptation can result in AI systems that operate under old assumptions, particularly problematic in domains such as healthcare, justice, or safety where the consequences of misalignment are significant. For example, notions of inclusivity may shift due to cultural changes, legal developments, or new ethical insights. An AI model trained on feedback from several years ago may not align with today's standards, undermining trust and effectiveness. This discrepancy highlights the necessity for AI systems to be adaptable and responsive to new information and changing contexts.

Moreover, the context-dependent aspect of human feedback adds another layer of complexity. Feedback that is appropriate and useful in one setting may be irrelevant or even misleading in another (Herel et al., 2024; Shen et al., 2024a). For instance, societal attitudes toward risk, autonomy, or harm mitigation can vary significantly between crisis scenarios and stable conditions. This variability means that AI systems trained to behave optimally under one set of conditions may fail to generalize effectively as the environment or societal norms evolve. Additionally, cultural differences can significantly influence perceptions and interactions, meaning that AI systems trained on data from one cultural background may not perform well or behave appropriately when deployed in a different cultural context (Yuan et al., 2024a; Feng et al., 2024). This inconsistency adds complexity to the alignment process, as systems must dynamically account for changes while avoiding overfitting to short-term or localized trends.

## 4.2. Challenges with AI Feedback

AI-based feedback mechanisms, using reward models or aligned AI systems to generate feedback, present unique challenges that can significantly impact the training and performance of alignment.

**Challenge: Dependence on underlying models.** A significant challenge of using AI-based feedback is its inherent dependence on the underlying models. These models, pivotal in determining the nature and quality of feedback, are typically trained on specific datasets, which might contain biases or represent only a subset of the broader population (Kirk et al., 2024; Santurkar et al., 2023). Consequently, the feedback generated by these models is limited by the diversity and the inherent characteristics of their training data. If the training data is skewed or non-representative, the AI can perpetuate or even amplify these biases in its feedback, leading to outcomes that may be misaligned with wider societal values or fail to accommodate the needs of all users. Furthermore, the models' dependence on their initial training constrains them to the specific scenarios and examples they were exposed to, limiting their adaptability

and applicability to new or evolving situations (Imran et al., 2025). This dependence underscores the need for ongoing updates and checks on the training data and model assumptions to ensure that AI-based feedback remains relevant.

**Challenge: AI-based feedback may not truly reflect human values.** Another challenge with AI-based feedback is that it may not accurately reflect human values, which is crucial for ensuring that AI systems behave in ways that are ethically and culturally appropriate. AI models, by their very nature, are limited to the data on which they are trained. These data, if not carefully curated, may lack the full spectrum of human experiences and moral considerations, leading to feedback that is biased or misaligned with societal expectations. Notably, studies have indicated that AI-based feedback suffers from several kinds of bias, including presentation-related biases (Shi et al., 2024; Zheng et al., 2023), social-related biases (Zhao et al., 2023b; Stammbach et al., 2024), content-related biases (Jiang et al., 2024), and cognitive-related biases (Panickssery et al., 2024). Beyond biases, LLM judges could also suffer from hallucination (Xu et al., 2024), making their judgments not based on facts and the human preferences they learned.

Moreover, LLM judges or reward models may optimize for quantifiable metrics such as accuracy that do not capture the nuances of human ethics or the complexities of moral reasoning (Li et al., 2025). This disconnect can result in AI behaviors that, while technically correct, are perceived as unethical, insensitive, or inappropriate by human users. Addressing this challenge involves not only diversifying the data used for training AI models but also integrating ethical oversight and human judgment into the loop of AI feedback generation, ensuring that AI systems continuously align with evolving human values and standards.

**Challenge: AI feedback lacks consistency and requires verification.** Ensuring the consistency of AI-generated feedback across similar data instances poses a significant challenge in data-centric alignment. Due to the stochastic nature of LLMs, the feedback provided for similar inputs can sometimes vary unpredictably, or suffer from position bias (Wang et al., 2024c). Researchers observed inconsistencies in GPT-4-based feedback, especially when two responses are subtly different (Tao & Li, 2025). For instance, when tasked with choosing the better response out of two nuanced options, GPT-4's selections can be evenly split among multiple trials, choosing each option about half the time. This variability can undermine the reliability of AI systems, especially when consistent responses are crucial for user trust and system integrity. Moreover, verifying the accuracy and appropriateness of AI-generated feedback adds another layer of complexity. Without human oversight, determining whether feedback is correct or suitably aligned

with intended outcomes is challenging. The challenge is compounded by the need for these verification systems to be transparent and understandable to human overseers (Liu et al., 2024c), ensuring that interventions can be made when discrepancies or errors in the feedback are detected.

# 5. Future Directions

## 5.1. Better Data Collection Practice

**Direction 1: Holistic feedback data collection.** To develop AI systems that accurately reflect global diversity and adapt to dynamic human contexts, a holistic feedback data collection approach is crucial. This involves diversifying data sources (*e.g.*, selected LLMs and annotators, as suggested by Kirk et al. (2024)) to encompass a broad spectrum of demographics, cultures, and environments, thereby ensuring no group is marginalized. Furthermore, the diversity of prompts used in data collection is critical. Incorporating a wider range of culturally and demographically diverse prompts can significantly enrich the diversity of scenarios and questions, ensuring that the data collection mirrors the multifaceted nature of human experiences.

To help community collect holistic feedback in practice, we suggest some criteria for evaluating the diversity of preference data. Specifically, we recommend considering diversity in three aspects: (1) **Annotator diversity**, which can be controlled by experimental design and has been explored in previous research (e.g., Santurkar et al. (2023), Kirk et al. (2024)) and HCI literature. Criteria including age, gender, ethnicity, country, religion, education, politology, and income, are usually considered. (2) **Prompt diversity**, referring to diversity in topic, task, and associated human values, helping AI align with human needs across scenarios. Within this aspect, topic diversity can be assessed via clustering (Kirk et al., 2024), while fundamental human values can evaluate value diversity (Shen et al., 2024a). Furthermore, prompt diversity can also be enhanced by the design of annotation tasks, which is common in other data collection settings (e.g., Yeh et al. (2024b) randomly assigns topics and instructions to collect logical fallacy data). (3) **Response diversity**, which captures the range of possible answers to a prompt. Sampling responses only from high-probability areas or a single model can reduce generalizability. To improve diversity, responses should be sampled from multiple models and different decoding strategies (Kirk et al., 2024).

**Direction 2: Toward dynamic and longitudinal preference collection.** Tracking the temporal drift of human preferences is a challenging and resource-intensive task, particularly when it is unclear which human values to prioritize and how frequently those values evolve. Understanding the dynamics of preference shifts requires a structured approach that balances feasibility and accuracy. The first step

is to investigate historical patterns of human value changes, identifying which values tend to fluctuate over time and the factors driving these changes. Techniques such as sentiment analysis on social media content, trend analysis in news articles, or longitudinal studies of public opinion can provide valuable insights into evolving societal norms and priorities. For example, Andrighetto & Vriens (2022) reviewed existing measurement techniques that assess the existence of social norms and tipping point models that understand norm change; while Lin et al. (2013) proposed to track opinion shifts in social media through sentiment analysis. These methods help pinpoint areas where alignment efforts should be focused and establish a baseline for monitoring future drift. Besides, Shen et al. (2024a) proposed a list of fundamental human values, which can serve as candidate values that need to be tracked over time.

Additionally, feedback-collection mechanisms must be designed to capture preferences at regular intervals, ensuring that systems remain responsive to the latest developments in human values. To make preference collection sustainable and efficient, it is crucial to integrate dynamic sampling techniques that focus on the most significant or rapidly changing values, reducing the burden of collecting and processing feedback for all potential preferences. Furthermore, the timing and frequency of feedback collection should be guided by data-driven insights, such as identifying periods of rapid societal change or emerging trends that are likely to impact alignment objectives. Another way to track changes in human values is to collect preferences for a fixed set of prompts regularly, a common practice in other fields like TREC (information retrieval) and SemEval (semantic analysis). The list of fundamental human values introduced by Shen et al. (2024a) can guide prompt construction.

Apart from changes in human values, there could be time-sensitive questions, which have varying answers based on timing (Herel et al., 2024). These questions often relate to politics, laws, and technology. To make LLM be aware of the impacts of time in answering questions, a time-sensitive preference dataset should include prompts covering these topics with different time frames.

**Direction 3: Deeper analysis should be done to validate the data collection protocols.** Our qualitative analysis of the existing alignment dataset demonstrated the benefits of incorporating "both are good" and "both are bad" options in preference labeling tasks, in addition to standard binary options. The additional categories allowed expressing genuine opinions, rather than forcing a choice when it was difficult to determine a preference. This enhanced protocol not only deepens our understanding of the feedback but also improves the reliability of human feedback. This finding indicates that many aspects of preference collection settings—such as the format of feedback, the variety

of response options, and the design of the data collection interface—remain under-explored and potentially influential in shaping the quality of feedback collected. We thus call for a comprehensive analysis of these factors to develop improved methods for collecting reliable feedback that accurately reflects human values. Note that some of the data collection issues, such as the options of "both are good" and "both are bad," have been studied in the social science field (Olsen, 1999). We believe that such findings in social science can serve as a guideline for developing a reliable preference collection protocol. Additionally, changes in feedback labeling can further influence the design of reward modeling and alignment algorithms. This highlights how focusing on data can drive algorithmic advancements, emphasizing the need for collaboration between data-centric and algorithm-centric approaches.

### 5.2. Data Cleaning Methods

**Direction 4: Human and AI collaboration to reduce unreliability.** In our exploration of the challenges associated with human feedback detailed in Section 4.1, human error emerged as a significant source of unreliability. To address the unreliability issue of human feedback, one straightforward approach is to clean up those unreliable data. For example, Wang et al. (2024a) proposed measuring the reward gap for each datum with a reward model, and flipping some data according to the reward gaps. Others have employed well-aligned LLMs, *e.g.*, GPT-4, as a judge to replace human annotators (Zheng et al., 2023). However, this introduces risks related to bias and the absence of human oversight, as discussed in Section 4.2. Our findings suggest that a collaborative approach, where human annotators work in tandem with AI through a committee of reward models, can effectively identify and correct errors. For example, Yeh et al. (2024a) find that human mistakes can be identified when a committee of reward models disagrees with original human labels. This led to the development of a data-cleaning process that adjusts data based on the committee's feedback, such as filtering data with low agreement. Models trained on the cleaned dataset significantly outperformed those trained on the original data. This exemplifies that a collaborative framework to refine feedback data is a promising direction.

**Direction 5: Prioritizing data quality over data size.** As highlighted in Section 4.1, the quality of prompts and responses in a feedback dataset is just as critical as the accuracy of feedback annotations. To ensure the feedback data genuinely reflects human values and preferences, it is essential to cleanse the dataset of not only flawed annotations but also unqualified prompts and responses. Recent studies underscore the importance of holistic data quality: Wu et al. (2024) demonstrated that adjusting the parameter $\beta$ in DPO can act as a filter to remove less informative samples, thereby enhancing the performance of the aligned

model. Li et al. (2024c) introduced Instruction Following Difficulty score that measures the degree to which given instruction benefits the alignment of the corresponding response, and selected data according to the scores. They showed that models trained with only 5% of the original Alpaca data beat the Alpaca model trained with full data. Lu et al. (2024) assigned semantic tags for each prompt and filtered out prompts based on the complexity or diversity of tags. They showed that models trained on only 6K selected data outperform models trained on more than 50K data on MT-Bench (Zheng et al., 2023). In addition, Yu et al. (2025) found that low-quality prompts are likely to produce low-quality responses and responses with larger variance. Hence, they proposed to filter prompts based on the rejected response reward and reward gap. Similarly, Shen et al. (2024b) found that a smaller, more task-relevant dataset could outperform a larger but less relevant one, indicating that solely increasing dataset size might even impair performance, particularly for out-of-domain tasks. These insights echo our direction 1 that instead of purely increasing dataset size, the diversity of data should be considered. Furthermore, these studies confirm the necessity of a comprehensive data-cleaning approach that considers the entire instance to maintain high data quality and relevance.

### 5.3. Feedback Verification and Oversight

**Direction 6: Incorporating human oversight for AI feedback.** Although AI can generate feedback at scale, its ability to discern nuanced human values and ethics without oversight is limited. Incorporating human oversight into AI feedback systems is essential for ensuring the ethical alignment and accuracy of AI operations. Human oversight involves systematic reviews and adjustments of AI-generated feedback by trained professionals who can assess the cultural, ethical, and contextual appropriateness of the AI's output. By establishing protocols for diverse human reviewers to regularly audit and refine AI feedback, organizations can maintain a balance between automation advantages and the indispensable insights that only human judgment can provide. This hybrid approach leverages the efficiency of AI while mitigating risks associated with biases or misinterpretations that AI systems may propagate, fostering more accurate and trustworthy AI feedback systems.

**Direction 7: Standardizing feedback verification.** To ensure the reliability and accuracy of AI systems, standardizing the verification of feedback within data-centric AI models is crucial. This process involves establishing consistent protocols and benchmarks for evaluating the quality and integrity of both human-generated and AI-generated feedback, while also ensuring such standardization is tailored to the specific deployment environments of the models, as different organizations may have varying use cases and requirements. By implementing uniform standards, orga-

nizations can systematically assess whether the feedback aligns with predefined criteria, such as relevance, objectivity, and completeness. This standardization not only aids in maintaining consistency across different stages of data processing and model training, but also allows developers and researchers to identify discrepancies and biases in feedback more effectively. The development of automated tools for continuous monitoring and validation of feedback further supports these efforts, facilitating the iterative improvement of alignment in response to dynamic and evolving feedback environments.

## 6. Conclusion

In this paper, we highlight several overlooked challenges of data-centric AI alignment. Through a qualitative analysis and comprehensive literature review, we found that although current feedback-collecting practices can collect large amounts of diverse feedback, human feedback often suffers from various sources of unreliability. Additionally, such feedback can be compromised by temporal changes and context-dependent variables, as well as by the dynamics of user interactions. We also revealed that AI-generated feedback might not accurately represent human values due to limitations inherent in the models used and the complex nature of human preferences.

Building on these insights, we propose several future directions for improving data-centric alignment. These include improving feedback collection practices, developing robust data-cleaning methodologies, and implementing rigorous feedback verification processes. We call for further research into these critical areas to ensure that feedback not only captures a wide range of human values, preferences, and goals but also does so accurately and reliably. Tackling these challenges is crucial for the development of AI systems that are effectively and meaningfully aligned with human intentions, enhancing their benefits in real-world applications.

## Impact Statement

Ensuring AI systems are aligned with human preferences is critical as these systems become more powerful and integrated into society. This position paper addresses overlooked challenges in data-centric AI alignment, emphasizing the importance of high-quality, reliable feedback in training and evaluation processes. By identifying key issues in human and AI-based feedback—such as unreliability, temporal drift, and context-dependence—and proposing actionable future research directions, this work aims to improve the alignment process, leading to AI systems that are safer, more ethical, and more responsive to evolving human needs. The insights and solutions offered here have the potential to shape the development of more trustworthy AI systems that positively impact individuals and society at large.

## Acknowledgement

We thank Hyeong Kyu Choi and Xuanming Zhang for their valuable suggestions on the draft. The authors would also like to thank the ICML anonymous reviewers for their helpful feedback. We gratefully acknowledge the support from the AFOSR Young Investigator Program under award number FA9550-23-1-0184, National Science Foundation Award No. IIS-2237037 & IIS-2331669, Office of Naval Research under grant number N00014-23-1-2643, Alfred P. Sloan Fellowship, and Philanthropic Fund from Schmidt Sciences Foundation and SFF.

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

# Position: Challenges and Future Directions of Data-Centric AI Alignment

Appendix

## A. Details of the Qualitative Analysis

### A.1. Data Annotation

**Annotation setup.** We randomly sample 80 data points from both harmless split and helpful split of Anthropic-HH dataset, and hire three annotators to re-label these 160 samples and record their thoughts and criteria during the annotation process. The three annotators are members of our lab who have great experiences in RLHF and LLM, and are familiar with the Anthropic-HH dataset. During our qualitative analysis, we label each data into one of the four categories: (1) chosen is better, (2) rejected is better, (3) both are good, and (4) both are bad. The additional categories of both are good and both are bad allowed expressing genuine opinions, rather than forcing a choice when it was difficult to determine a preference. Apart from this, we follow the annotation protocol of Anthropic-HH to define "better" in terms of harmlessness and helpfulness by themselves, allowing them to use their subjective judgment. This flexibility allows annotators to consider context, personal experience, and the specific nuances of each data point, ultimately reflecting real-world ambiguities and diversity in human feedback. After collecting the annotations, we employ majority voting to determine the final preference label of human annotators for each data point. For cases where the three annotators chose three different labels, we mark the data as having an "uncertain" preference label.

**Annotation result.** We observe that less than 5% of the data are labeled as uncertain, and 25% of them are "rejected is better," contradicting the original label. In addition, another 25% of samples are labeled as "both are bad," suggesting a poor quality of response candidates. The Fleiss's $\kappa$ inter-annotator agreement is 0.46, which is at the level of moderate agreement.

### A.2. Identifying Sources of Unreliability in Human Feedback

Based on the annotations, we conducted a thematic analysis to identify the sources of unreliability in the human feedback dataset. Specifically, we focused on samples labeled as "rejected is better" and "both are bad," as well as those with low inter-annotator agreement. We reviewed these samples along with the notes provided by annotators, creating a list of codes to represent the observed concepts and attributes associated with the reasons for unreliability. Each sample was assigned a code. After completing this coding process, we grouped related codes into six overarching themes, which we refer to as the sources of unreliability. These themes and their corresponding codes are as follows:

- Source 1: Mistakes made by human
    - Identifiable mistakes: Annotators explicitly note errors in the original label.
    - Blank notes: Annotators provide no notes for the sample (usually because the mistakes are obvious).
- Source 2: High subjectivity and lack of context
    - Recommendation prompts: User queries involve recommendations (*e.g.*, travel, books, music).
    - Greetings/conversational closings: Both responses are greetings or conversation-ending phrases.
    - Open-ended prompts: Prompts that lack specific goals or criteria, leading to subjective interpretations.
- Source 3: Different preference criteria
    - Concise vs. detailed responses: One response is concise, while the other is detailed.
    - Direct (but imprecise) answer vs. clarified question: One response answers directly but lacks precision, while the other clarifies the query before providing an answer.
    - Formal vs. casual tone: Annotators disagree based on tone preference (e.g., professional vs. conversational).
    - Different types of flaws: Annotators highlight different flaws in the responses (e.g., one is incomplete, the other contains misinformation).
- Source 4: Different threshold of criteria
    - Harmfulness level: Annotators disagree on the degree of harmfulness in the responses.
    - Tolerance for off-topic content: Annotators vary in their tolerance for off-topic elements.
    - Potential harm: Annotators differ in their perception of the potential harmfulness of a response. Grammar/spelling errors: Disagreement on the importance of minor linguistic errors.
- Source 5: Harmful suggestions in both responses
    - Toxic language: Both responses include toxic language.

- Encouragement of illegal actions: Both responses advocate for illegal behavior.
- Encouragement of risky behaviors: Both responses promote actions that are unsafe or dangerous, even if not explicitly illegal.
- Source 6: Misinformation/irrelevant in both responses
  - Hallucination: Both responses include fabricated or nonsensical information.
  - Misinformation: Both responses contain incorrect information.
  - Off-topic content: Both responses fail to address the prompt coherently.

