# OpenReview forum: "Position: Challenges and Future Directions of Data-Centric AI Alignment"
_ICML.cc/2025/Position_Paper_Track — ICML 2025 Position Paper Track poster_

### Official Review · Reviewer_4SBL · 2025-02-25

**Significance:** 3
**Argument Clarity:** 2
**Rating:** 3
**Confidence:** 3

**Questions:**

- For the future directions, especially regarding data collection more concrete ideas about how the community should proceed?

**Discussion Potential:**

2

**Paper Summary:**

This paper presents the importance of data-centric AI alignment, focusing on challenges and future directions. It argues that current AI alignment approaches often emphasize on algorithmic techniques like RLHF, while underestimating the critical role of data quality and representativeness. The authors highlight challenges related to two categories of alignment, human feedback and AI-generated feedback, including issues of reliability, temporal drift, context dependence, and model limitations. They identified multiple sources of unreliability, and propose future research directions to improve feedback collection, data cleaning, and verification processes. The paper advocates for a shift towards data-centric approaches to enhance the alignment of AI systems with human values.

## update after rebuttal

I have no more questions, and I will keep my score.

**Position:**

Yes

**Position In Title:**

Yes

**Related Work:**

2

**Strengths And Weaknesses:**

Strengths

- The topic is timely and relevant. AI alignment is a crucial concern as AI systems become more powerful. The paper raises important concerns about the relatively overlooked role of data-centric approaches.
- The paper presents a detailed and systematic breakdown of the limitations of both human feedback and AI-based feedback. The identification of key sources of unreliability in human feedback is particularly useful.
- The arguments is well-supported with analysis and related work review. Future research directions are appealing and interesting.

Weaknesses

- The quantitative validation is limited while the qualitative analysis is insightful. The paper lacks quantitative measure of bias and unreliability in human feedback datasets to further strengthen the claims.
- Some future directions are a bit too broad and challenging, for example, holistic data collection across demographics cultures and environments and dynamically tracking the trend of preferences. These are not only data problems for alignment, but problems for all kinds of data and their usage in ML. From the paper there is no concrete ways of solving them.

**Support:**

2

---

> ### Author Rebuttal · Authors · 2025-03-26
>
> We sincerely appreciate your feedback and insightful comments! We address the questions and comments below in detail.
>
> > Quantitative measure of bias and unreliability in human feedback datasets
>
> Thank you for this excellent suggestion. In addition to the qualitative insights, we have conducted quantitative analyses, detailed in **Appendix L722–L740**, to empirically ground our observations.
>
> Specifically, we re-annotated 160 samples from the HH-RLHF dataset, each with three independent annotations. Strikingly, in **25%** of the samples, annotators judged the “rejected” response to be better than the “chosen” one, directly contradicting the original labels—underscoring serious quality concerns in the dataset. Furthermore, another 25% of the samples were labeled “both are bad,” suggesting that the candidate responses themselves were often low quality, leading to unreliable supervision signals.
>
> To gain deeper insight into the nature of annotation noise, we further analyzed the subset of samples with low inter-annotator agreement (IAA). As shown in Table 1 below, a substantial portion of disagreement stems from subjectivity (28%), different preference criteria (29%), and varying thresholds of criteria (37%).
>
> We also analyzed the samples where annotators agreed that both responses were bad. As shown in Table 2 below, most of the unreliablility stems from different preference criteria (25%) and harmful suggestion (39%) and misinformation (36%) in both responses. In contrast to the IAA case, these samples reflect more clearly that poor response quality—rather than annotator—can trigger unreliable supervision signals.
>
> Together, these findings provide strong quantitative support for the challenges we raise in the main paper.
>
> **Table 1. Proportion of sources of unreliability in "low IAA" data**
>
> |Source of unreliability| percentage|
> |---|---|
> |Mis-labeling by human|2%|
> |High subjective query|28%|
> |Different preference criteria|29%|
> |Different thresholds of criteria|37%|
> |Harmful suggestions in both responses|0%|
> |Misinformation/irrelevant in both responses|4%|
>
> **Table 2. Proportion of sources of unreliability in "both are bad" data**
>
>
> |Source of unreliability| percentage|
> |---|---|
> |Mis-labeling by human|0%|
> |High subjective query|0%|
> |Different preference criteria|25%|
> |Different thresholds of criteria|0%|
> |Harmful suggestions in both responses|39%|
> |Misinformation/irrelevant in both responses|36%|
>
>
> > Future directions 1 and 2 are a bit too broad and challenging
>
> Thank you for this thoughtful observation. We agree that challenges such as holistic data collection across demographics, cultures, and environments, as well as dynamically tracking preference trends, are broad and foundational issues that extend beyond alignment alone. Our goal in highlighting these directions is not to claim that they are unique to alignment, but rather to emphasize that alignment—particularly in the context of LLMs—magnifies these challenges due to the inherently subjective, contextual, and evolving nature of human preferences.
>
> While as a position paper, we do not claim to offer solutions, we see value in articulating these open problems to encourage more principled, data-centric approaches to alignment research. In the revised version, we will clarify that certain benchmarking efforts (e.g., tracking how preferences shift over time and measuring stability of preferences across rounds) offer starting points for operationalizing these goals, not definitive answers. **We hope this framing better positions our work as a call to action for the community to collectively advance in these complex but crucial directions.**

---

> > ### Comment · Reviewer_4SBL · 2025-04-02
> >
> > Thank the authors for their response. I have no more questions, and I will keep my score.

---

> > > ### Author Response · Authors · 2025-04-03
> > >
> > > We are glad the response addressed your concerns, and once again, thank you for your recognition of our work!

---

### Official Review · Reviewer_zjJT · 2025-03-08

**Significance:** 4
**Argument Clarity:** 3
**Rating:** 4
**Confidence:** 3

**Questions:**

None

**Discussion Potential:**

4

**Paper Summary:**

The paper argures for data-centric AI alginment setting forth a number of challenges and future directions.

## update after rebuttal
Reviews and rebuttal read. Thank you. No update was necessary.

**Position:**

Yes

**Position In Title:**

Yes

**Related Work:**

3

**Strengths And Weaknesses:**

* The problem is highly timely and relevant for the ML community.
* While the position itself is not novel - the last years have seen a continuous rise in the relevance of data, the paper has significant depth to make it important.
* The paper sets forth a number of concrete examples making it very tangible while at the same time also highlighting general principles and comparison among data / algorithmic-centric alignment.
* The paper's scope is mostly on LLMs, which is ok, but could be more clealry articulated.

**Support:**

3

---

> ### Author Rebuttal · Authors · 2025-03-26
>
> We sincerely appreciate your positive feedback and insightful comments! We address the questions and comments below in detail.
>
> > "_The paper's scope is mostly on LLMs, which is ok, but could be more clearly articulated."_
>
> Thank you for this suggestion! We will clarify this in the revision by explicitly stating in the 4-th paragraph of the introduction: "In this work, we primarily focus on the challenges of aligning LLMs." We believe this focus is timely and appropriate given that LLMs are currently the dominant foundation for many AI systems deployed in real-world applications. They are also the main targets of recent alignment efforts using human and AI feedback.

---

### Official Review · Reviewer_wFqk · 2025-03-10

**Significance:** 2
**Argument Clarity:** 3
**Rating:** 3
**Confidence:** 4

**Questions:**

- i guess there should be some connections between data attribution methods between direction 1,4,5? there are various papers in that field already digging into improving the overall quality of dataset for alignment

**Discussion Potential:**

3

**Paper Summary:**

The paper argues that AI alignment research should put more emphasis on algorithmic-centric methods to a more data-centric approach, which concentrates on the quality and representativeness of feedback. The authors present a comprehensive qualitative analysis o the challenges associated with both human/AI-based feedback. And they also outline several future research directions including improved data collection, data cleaning and better verification process.

**Position:**

Yes

**Position In Title:**

No

**Related Work:**

3

**Strengths And Weaknesses:**

* Strength
  * the raised issue is rather important and timely
  * many actionable future directions, which are definitely beneficial for the community to follow-up
  * broad literature survey

* Weakness
  * I don't feel like some of these challenges are relatively newly identified. In the literature of data selection/data cleaning, there are already many evidence that those human preference dataset can be improved/filtered/selected.
  * Are the sources you identified really matter? Is there any quantitive numbers to indicate that these alignment datasets really face these challenges?
  * weak coverage on current attempt of the data-centric alignment papers

**Support:**

3

---

> ### Author Rebuttal · Authors · 2025-03-26
>
> We sincerely appreciate your feedback and insightful comments! We address the questions and comments below in detail.
>
> > Challenges of data selection/data cleaning are not relatively newly identified.
>
> We agree that data selection has been a long-standing topic in machine learning, and we will add citations in Section 5.2 to acknowledge this. However, our paper aims to go beyond standard data cleaning. We offer a holistic perspective on data-centric challenges for alignment, covering not only filtering but also issues of data curation, selection, verification, and evaluation, all in the specific context of AI alignment. The alignment setting introduces new complexities—e.g., subjective preferences, evolving goals, and feedback ambiguity—which call for a renewed and focused lens on these data challenges.
>
> > Are there any quantitive numbers to indicate that these alignment datasets really face these challenges?
>
> Yes. We have conducted a detailed quantitative analysis, presented in **Appendix L722–L740**. These analyses complement our qualitative insights and empirically substantiate the prevalence and types of unreliability present in real-world preference datasets. We kindly refer you to our response to Reviewer 4SBL (R4) for a fuller discussion.
>
> > Weak coverage of the current attempt of the data-centric alignment papers
>
> Thank you for pointing this out. We already cite a broad range of papers and welcome any specific suggestions for additional ones. Below is a consolidated list of works we currently cite and plan to add in the revision:
>
> 1. Wang, Bing et al. "Secrets of RLHF in Large Language Models Part II: Reward Modeling." arXiv preprint (2024).
> 2. Ryan, Michael J et al. "Unintended Impacts of LLM Alignment on Global Representation." ACL (2024).
> 3. Santurkar, Shibani et al. "Whose Opinions Do Language Models Reflect?" ICML (2023).
> 4. Lerner, Maria et al. "Whose Preferences? Differences in Fairness Preferences and Their Impact on the Fairness of AI Utilizing Human Feedback." ACL (2024).
> 5. Kirk, Hannah Rose et al. "The PRISM Alignment Dataset: What Participatory, Representative and Individualised Human Feedback Reveals About the Subjective and Multicultural Alignment of Large Language Models." NIPS (2024).
> 6. Zheng, Lianmin et al. "Judging LLM-as-a-Judge with MT-Bench and Chatbot Arena." NIPS (2023)
> 7. Lee, Harrison et al. "RLAIF vs. RLHF: Scaling Reinforcement Learning from Human Feedback with AI Feedback." ICML (2024).
> 8. Cui, Ganqu et al. "UltraFeedback: Boosting Language Models with High-quality Feedback." arXiv (2023).
> 9. Li, Haitao et al. "LLMs-as-Judges: A Comprehensive Survey on LLM-based Evaluation Methods." arXiv (2024).
> 10. Li, Dawei et al. "From Generation to Judgment: Opportunities and Challenges of LLM-as-a-judge." arXiv (2024).
> 11. Lee, JoonHo et al. "Improving Instruction Following in Language Models through Proxy-Based Uncertainty Estimation." ICML (2024).
> 12. Bai, Yuntao et al. "Training a Helpful and Harmless Assistant with Reinforcement Learning from Human Feedback." arXiv (2022)
> 13. Shen, Hua et al. "ValueCompass: A Framework of Fundamental Values for Human-AI Alignment." ArXiv (2024).
> 14. Shi, Lin et al. "Judging the Judges: A Systematic Investigation of Position Bias in Pairwise Comparative Assessments by LLMs." arXiv (2024).
> 15. Zhao, Yachao et al. "Mind vs. Mouth: On Measuring Re-judge Inconsistency of Social Bias in Large Language Models." arXiv (2023).
> 16. Panickssery, Arjun et al. "LLM Evaluators Recognize and Favor Their Own Generations." NIPS (2024).
> 17. Shen, Judy Hanwen et al. "Towards Data-Centric RLHF: Simple Metrics for Preference Dataset Comparison." arXiv (2024).
> 18. Yu, Ping et al. "R.I.P.: Better Models by Survival of the Fittest Prompts." ArXiv (2025).
> 19. Li, Ming et al. "From Quantity to Quality: Boosting LLM Performance with Self-Guided Data Selection for Instruction Tuning." NAACL (2024).
> 20. Khaki, Saeed et al. "RS-DPO: A Hybrid Rejection Sampling and Direct Preference Optimization Method for Alignment of Large Language Models." Findings of NAACL (2024).
>
> > Q: Are there connections between data attribution methods between directions 1,4,5 in the literature?
>
> This is an insightful question. We did find papers addressing directions 4 and 5, such as the references we listed below. However, to our best knowledge, very few papers explicitly connect these stages.  Bridging these stages remains an open challenge. We will clarify this connection and emphasize it as a promising direction for future work in our revised draft.
>
> [1]: Yu, Ping et al. "R.I.P.: Better Models by Survival of the Fittest Prompts." ArXiv (2025).
>
> [2]: Li, Ming et al. "From Quantity to Quality: Boosting LLM Performance with Self-Guided Data Selection for Instruction Tuning." NAACL (2024).
>
> [3]: Khaki, Saeed et al. "RS-DPO: A Hybrid Rejection Sampling and Direct Preference Optimization Method for Alignment of Large Language Models." Findings of NAACL (2024).

---

> > ### Comment · Reviewer_wFqk · 2025-04-05
> >
> > Thank you for the response. I have updated my score.

---

> > > ### Author Response · Authors · 2025-04-07
> > >
> > > We are glad the response addressed your concerns, and once again, thank you for your recognition of our work!

---

### Official Review · Reviewer_zW7Y · 2025-03-14

**Significance:** 4
**Argument Clarity:** 4
**Rating:** 4
**Confidence:** 4

**Questions:**

1. How much do the challenges / flaws in human and AI generated feedback effect the results of the trained models? Are studies showing this quantitatively? It looks like first empirical papers came out on this (see Eckman et al. 2025 on the PAIR framework ... preprint .. for what is on my mind when I ask this question).

2. What criteria would you envision to be used in 5.1 and 5.2 future directions? Is it just prompts in the human feedback settings or are other data envisioned?

**Discussion Potential:**

3

**Paper Summary:**

The paper makes the important point that AI systems need to be trained on / based on data that properly reflect the real world. It raises concerns that self-labelling of data through AI systems lead to biases (inherent in the trained data). It discusses human vs. AI feedback. The paper has a long intro section that explains the difference between human feedback and AI-generated feedback, as well as the human vs. algorithmic centric alginment.  It has one section that goes through examples for challenges with both human and ai-generated feedback. The final section is a suggestion on how feedback could be aquired in the future.

**Position:**

Yes

**Position In Title:**

Yes

**Related Work:**

2

**Strengths And Weaknesses:**

The topic is super important and fits the conference.

The paper cites an enourmous amount of papers, which is a great resource, but given that in support of some sections a long list of authors is listed, a reader would often not know why and how the cited references support the claims.

A more careful use of references, or an explanation of what each of these references support/show/demonstrated would be much more useful.

The qualitative discussion of challenges is very thought provoking. The readers would be more convinced with studies showing the quantitative effects of the challenges.

Some of the challenges mentioned are well researched in the social sciences. Source 2 and 3 for example have been quantitatively sutdied here (https://aclanthology.org/2024.uncertainlp-1.8.pdf) and many more papers exist in the larger social science field to support these (very valid) intuitions. Take for example line 255-264 (source 5 but also source 6) is an effect known in the social science, (see here for pointers https://www.sciencedirect.com/science/article/abs/pii/S095032939900049X) and  many of these studies show quantitative results. Now granted, those may or may not translate to the RHLF situations, a point that can be discussed.

The directions for future research are interesting and could be fleshed out more. There are three categories: Data collection, data cleaning , and feedback. I like that there is acknowledgement of the longitudinal nature and the data and the need to take that into account. Both for the holistic feedback and the logitudinal the paper would be stronger if criteria would be mentioned how this can be evaluated. Defining the population/space that needs to be represented and defining when is good.

A valuable line of thinking could be the one of Groves and others showing when biases in statistical data collection occure, and their emphasis that such bias only arises if the outcome of interest (here the label) is indeed correlated with charateristics of the ones providing the ratings. Implicitly this is clear, in section 5 it could be discussed more specifically. It is easy to say it should be diverse and include different cultures. For some tasks this might be the case for others not.

The data cleaning point direction 5 is also great and touches on a much larger point which is data quality vs. data size. I wonder if the heading could be changed to make that point stronger.

**Support:**

2

---

> ### Author Rebuttal · Authors · 2025-03-26
>
> We sincerely appreciate your feedback and insightful comments! We address the questions and comments below in detail.
>
> > Citation clarity
>
> Thank you for pointing this out. Our intention was to provide a comprehensive survey of the rapidly growing literature in AI alignment, to situate our position paper within the broader context and demonstrate the momentum and diversity of research.
>
> To improve readability, we will (1) restructuring some paragraphs to explain references by subtopic or contribution, and (2) selectively annotating key citations with brief clarifying remarks to better highlight their role in supporting each claim. This will better highlight their relevance while maintaining the paper’s value as a resource.
>
> > "_The qualitative discussion of challenges is very thought provoking. The readers would be more convinced with studies showing the quantitative effects of the challenges."_
>
> Thank you for this excellent suggestion. We provide detailed quantitative evidence in **Appendix L722–L740**, and kindly refer you to our response to R4 for further elaboration.
>
> > Connection to social science literature.
>
> We greatly appreciate this insightful comment. Indeed, many sources we identify have deep roots in the social sciences. We will add relevant citations [1,2] and contextualize our findings accordingly. We also agree with the reviewer’s excellent point that while these findings may not fully transfer to RLHF settings, exploring these parallels helps enrich the interdisciplinary foundation of alignment research.
>
> > Change heading of direction 5
>
> Thank you—will do. We will rename it to "Prioritizing data quality over data size."
>
> > Q1: How much do the challenges/flaws in human and AI-generated feedback affect the results of the trained models?
>
> Thank you for this important question. Recent work by [3] (cited in L368) offers quantitative evidence. In their Figure 4, reward models trained on incorrect preferences perform worse than random guessing, those trained on ambiguous data perform on par with guessing, and only strong preferences yield meaningful learning. This strongly supports our argument that feedback data quality has a critical impact on alignment.
>
> > Q2: Criteria for section 5.1 and 5.2
>
> Thank you for this important question. For 5.1, we define diversity in three aspects: (1) **Annotator diversity**, which can be controlled by experimental design and has been explored in previous research (e.g., [4, 5]) and HCI literature. Criteria include age, gender, ethnicity, country, religion, education, politology, and income. (2) **Prompt diversity**, referring to diversity in _topic_, _task_, and _associated human values_, helping AI align with human needs across scenarios. Topic diversity can be assessed via clustering [5], while fundamental human values [6] can evaluate value diversity. Furthermore, prompt diversity can be enhanced by design of annotation tasks (e.g., [7] randomly assigns topics and instructions). (3) **Response diversity**, which captures the range of possible answers to a prompt. Sampling responses only from high-probability areas or a single model can reduce generalizability. To improve diversity, responses should be sampled from multiple models [5] and different decoding strategies.
>
> For 5.2, we consider: (1) **Time-sensitive questions**, which have varying answers based on timing [8]. These often relate to politics, laws, and technology. A time-sensitive preference dataset should include prompts covering these topics with different time frames. (2) **Changes in human values**, where preferences evolve over time. To track this, preferences for a fixed set of prompts can be collected regularly (e.g., annually), a common practice in other fields like TREC (information retrieval) and SemEval (semantic analysis). The list of fundamental human values [6] can guide prompt construction.
>
> [1]: Beck, Jacob et al. "Order effects in annotation tasks: Further evidence of annotation sensitivity." UncertaiNLP (2024).
>
> [2]: Olsen, Svein Ottar. "Strength and conflicting valence in the measurement of food attitudes and preferences." Food Quality and Preference 10 (1999).
>
> [3]: Wang, Bing et al. "Secrets of RLHF in Large Language Models Part II: Reward Modeling." arXiv preprint (2024).
>
> [4]: Santurkar, Shibani et al. "Whose Opinions Do Language Models Reflect?" ICML (2023).
>
> [5]: Kirk, Hannah Rose et al. "The PRISM Alignment Dataset: What Participatory, Representative and Individualised Human Feedback Reveals About the Subjective and Multicultural Alignment of Large Language Models." NIPS (2024).
>
> [6]: Shen, Hua et al. "ValueCompass: A Framework of Fundamental Values for Human-AI Alignment." ArXiv preprint (2024).
>
> [7]: Yeh, Min-Hsuan et al. "CoCoLoFa: A Dataset of News Comments with Common Logical Fallacies Written by LLM-Assisted Crowds." EMNLP (2024).
>
> [8]: Herel, David et al. "Time Awareness in Large Language Models: Benchmarking Fact Recall Across Time." ArXiv preprint (2024).

---

> > ### Comment · Reviewer_zW7Y · 2025-04-07
> >
> > Thanks for providing the information. I updated my scores.

---

> > > ### Author Response · Authors · 2025-04-07
> > >
> > > We are glad the response addressed your concerns, and once again, thank you for your support and insightful comments!

---

### Decision · Program_Chairs · 2025-04-29

**Decision:**

Accept (poster)

**Comment:**

The paper addresses a timely and relevant topic within AI alignment and presents thorough evidence and actionable recommendations for the argued position.